# Indole-3-acetic acid synthesized through the indole-3-pyruvate pathway promotes *Candida tropicalis* biofilm formation

**Masaru Miyagi**[1]*, **Rachel Wilson**[2], **Daisuke Saigusa**[3], **Keiko Umeda**[3], **Reina Saijo**[3], **Christopher L. Hager**[2], **Yuejin Li**[2], **Thomas McCormick**[2], **Mahmoud A. Ghannoum**[2,4]*

**1** Department of Pharmacology, Case Western Reserve University, Cleveland, Ohio, United States of America, **2** Department of Dermatology, Center for Medical Mycology, Case Western Reserve University, Cleveland, Ohio, United States of America, **3** Department of Integrative Genomics, Tohoku Medical Megabank Organization, Tohoku University, Sendai, Japan, **4** University Hospitals Cleveland Medical Center, Cleveland, Ohio, United States of America

\* mxm356@case.edu (MM); mag3@case.edu (MAG)

**Data Availability Statement:** All relevant data are within the manuscript and its Supporting information files.

## Abstract

We previously found that the elevated abundance of the fungus *Candida tropicalis* is positively correlated with the bacteria *Escherichia coli* and *Serratia marcescens* in Crohn's disease patients and the three pathogens, when co-cultured, form a robust mixed-species biofilm. The finding suggests that these three pathogens communicate and promote biofilm formation, possibly through secretion of small signaling molecules. To identify candidate signaling molecules, we carried out a metabolomic analysis of the single-species and triple-species cultures of the three pathogens. This analysis identified 15 metabolites that were highly increased in the triple-species culture. One highly induced metabolite was indole-3-acetic acid (IAA), which has been shown to induce filamentation of certain fungi. We thus tested the effect of IAA on biofilm formation of *C. tropicalis* and demonstrated that IAA promotes biofilm formation of *C. tropicalis*. Then, we carried out isotope tracing experiments using $^{13}$C-labeled-tryptophan as a precursor to uncover the biosynthesis pathway of IAA in *C. tropicalis*. The results indicated that *C. tropicalis* synthesizes IAA through the indole-3-pyruvate pathway. Further studies using inhibitors of the indole-3-pyruvate pathway are warranted to decipher the mechanisms by which IAA influences biofilm formation.

## Introduction

Inflammatory bowel disease (IBD) is a relapsing-remitting systemic disorder that involves chronic inflammation of the gastrointestinal tract. The two main types of IBD are Crohn's disease (CD) and Ulcerative colitis (UC). CD may affect any area of the gastrointestinal tract, whereas UC primarily impacts the large intestine and the rectum. The 2015 National Health Interview Survey estimated that approximately 3 million US adults ($\geq$18 years) suffer from IBD [1]. The etiology of IBD is not entirely understood, however accumulating studies suggest that dysbiosis of the gut microbiome plays a pivotal role in triggering chronic inflammation associated with IBD [2–4]. Our recent study, which compared the gut bacteriome and mycobiome of CD patients to their co-habiting unaffected relatives, indicated that the abundance of

**Funding:** This work was supported in part by a National Institutes of Health Grant # R01AI145289-01A1 to M.A.G., the Tohoku Medical Megabank Project (Grant # JP20km0105001 and JP20km0105002) from the Ministry of Education, Culture, Sports, Science and Technology of Japan (MEXT) and Japan Agency for Medical Research and Development (AMED), and Project for Promoting Public Utilization of Advanced Research Infrastructure from MEXT.

**Competing interests:** The authors have declared that no competing interests exist.

the fungus *Candida tropicalis* (*C. tropicalis*) is significantly elevated in CD patients compared to unaffected relatives and abundance is positively correlated with the levels of two pathogenic bacteria; *Escherichia coli* (*E. coli*) and *Serratia marcescens* (*S. marcescens*) [5]. Our *ex-vivo* study also demonstrated that the mass and thickness of triple-species (*C. tropicalis* + *E. coli* + *S. marcescens*) biofilm are significantly greater than those of the single-species biofilms formed by each pathogen alone [5]. These results indicate that the three pathogens communicate with each other and facilitate the formation of polymicrobial biofilm. Thus, uncovering the underlying mechanism by which these pathogens form a robust polymicrobial biofilm is essential to understanding the pathogenicity of these organisms and may lead to developing a therapeutic approach for the treatment and prevention of CD-associated gastrointestinal inflammatory symptoms.

One of the means that microorganisms use to communicate with each other and to influence each other's behavior is through secretion of small signaling molecules [6, 7]. Some of these are known as quorum sensing molecules that enable microbes to sense cell population density and adjust gene expression accordingly [8]. For example, farnesol, tyrosol, 2-phenylethanol, and tryptophol are known quorum sensing molecules in fungi [9] and it is known that farnesol inhibits, while tyrosol promotes, biofilm formation [10, 11]. In addition to the quorum sensing molecules, a number of small signaling molecules are known to be secreted from fungi [12].

Indole-3-acetic acid (IAA), one of the molecules known to be secreted by fungi, has been reported to induce filamentation of certain fungi [13–16]. a vital pathogenicity factor of invasive fungal infection. It is known that L-tryptophan (Trp) is the primary precursor for IAA biosynthesis by microorganisms [17]. At least five Trp-dependent IAA biosynthesis pathways have been described (Fig 1) [18]. Among those, indole-3-acetamide or indole-3-pyruvate pathways have been indicated to be the predominant pathway used by fungi [14, 17]. However, prior studies of IAA biosynthesis were done mostly with fungi that interact with plants, and no study has been performed using *Candida* species which are responsible for the majority of fungal infections in humans.

In the present study, we first carried out metabolomic analysis of the single-species and triple-species cultures of the three pathogens (*C. tropicalis*, *E. coli*, and *S. marcescens*) to identify molecules that may promote biofilm formation. The initial analysis identified metabolites that were highly elevated in the triple-species culture compared to the individual single-species cultures. One of the identified metabolites was IAA. We then tested whether IAA has any effect on the formation of *C. tropicalis* biofilm using *in vitro* assays. Lastly, we investigated the biosynthesis pathway of IAA in *C. tropicalis* by isotope tracing analysis.

## Materials and methods

### Materials

$^{13}C_{11}$-Trp and $d_7$-indole-3-acetic acid ($d_7$-IAA) were purchased from Cambridge Isotope Laboratories (Tewksbury, MA). Trp, indole-3-lactic acid (ILA), indole-3-ethanol (tryptophol, TOL), indole-3-acetic acid (IAA), indole-3-acetonitrile (IAN), indole-3-acetamide (IAM), tryptamine (TAM) were from Sigma-Aldrich (St. Louis, MO). HPLC grade water and acetonitrile were obtained from Fisher Scientific (Pittsburgh, PA). All other commercially available chemicals were either reagent grade or were of the highest quality.

### Strains and culture media

*C. tropicalis* (CT), *E. coli* (EC), *and S. marcescens* (SM) used in this study were isolated from the fecal sample of a CD patient (MRL32707, MRL36101, MRL36102, respectively).

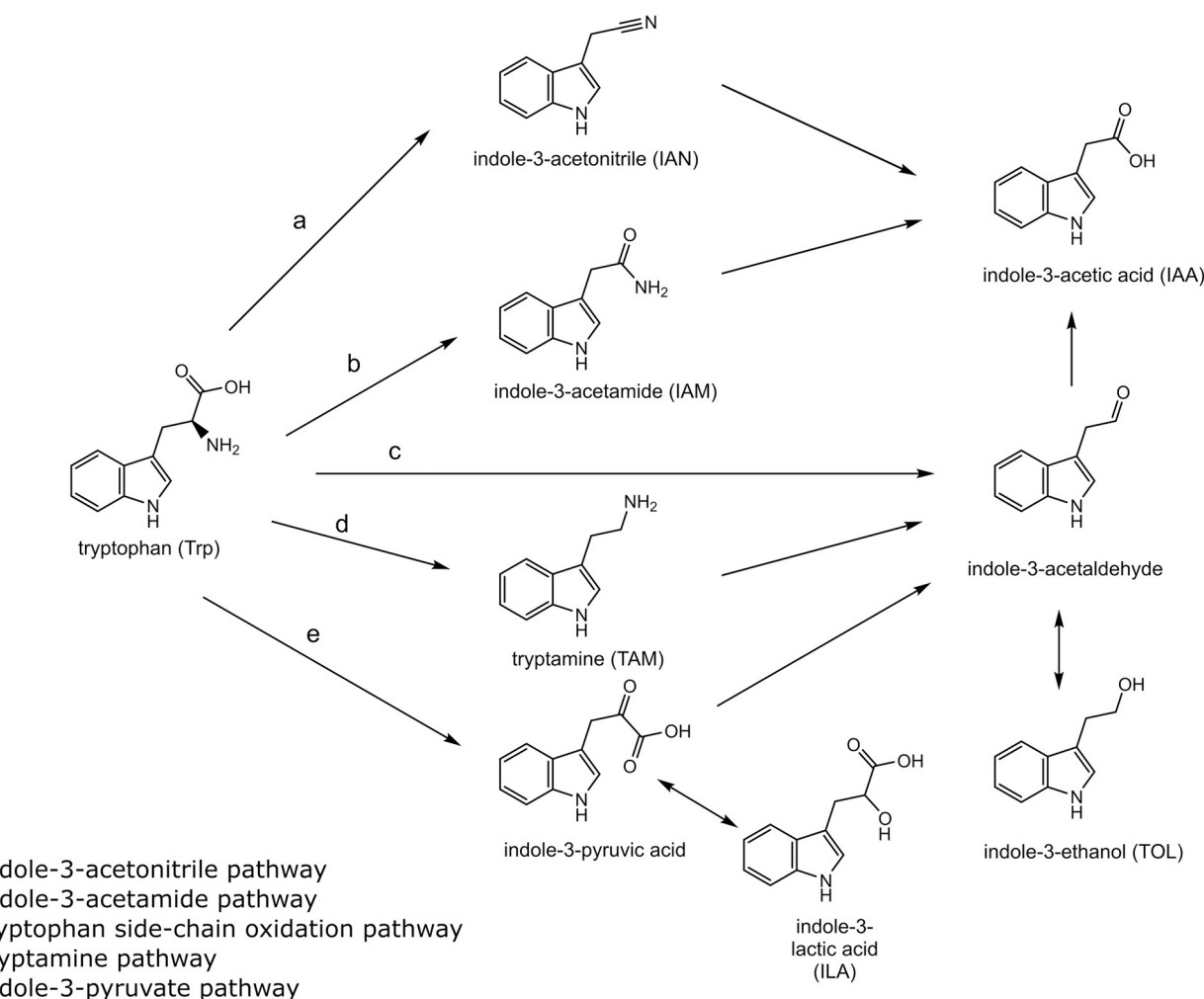

**Fig 1. Tryptophan-derived IAA biosynthesis pathways in bacteria.** Trp, IAN, IAM, TAM, ILA, TOL, and IAA were monitored. Indole-3-pyruvic acid and indole-3-acetaldehyde were not monitored in this study because they are not stable.

## Targeted metabolomics

Each of the clinical isolates were streaked on a potato dextrose agar (PDA) plate (Becton Dickinson, Sparks, MD) and incubated at 37°C for 16 h. One loopful of a colony of the microbes was then incubated in 10 mL of phosphate buffer saline (PBS) at 37°C for 16 h, and the cells were collected by centrifugation at 2,500 g for 5 min at 4°C. The cell pellet was then resuspended in RPMI 1640 (United States Biological, Salem, MA) to 1 x $10^7$ blastospore/mL.

Single species cultures were carried out by adding each of the primary biofilm inoculum of CT, EC, and SM (1 x $10^7$ cells per mL) into 40 mL of RPMI 1640 medium and incubated at 37°C for 24 h. Triple species cultures were carried out similarly with an equal number of CT, EC, and SM (0.33 x $10^7$ cells/mL each, keeping the final number of cells the same across all co-cultures). Following incubation, the culture media were collected, centrifuged at 3,000 x g for 5 min, then filtered using a 0.22 μm 50 mL vacuum filter tube (MilliporeSigma, Burlington, MA), and stored until analysis.

The metabolites in the culture media were analyzed as described previously [19]. Briefly, metabolites were extracted from 50 μL of culture medium by mixing with 250 μL of mixed

water:methanol:chloroform (1:2.5:1, v/v/v) containing 5 μg of 2-isopropylmalate (Sigma-Aldrich, Tokyo, Japan) as an internal standard. The solution was shaken at 1,200 rpm for 30 min at 37˚C and centrifuged at 16,400 x g for 10 min at 4˚C. The resulting supernatant was collected and dried in a Speed-vac concentrator. To the extracted metabolites, 80 μL of methoxyamin hydrochloride (Sigma-Aldrich, Tokyo, Japan) solution (20 mg/mL in pyridine) was added and incubated for 90 min at 30˚C for oximation (meto) of metabolites, and then 40 μL of *N*-methyl-*N*-trimethylsilyl-trifluoroacetamide (GL Science, Tokyo, Japan) was added and incubated for 30 min at 37˚C for trimethylsilylation (TMS). The resulting solution was centrifuged at 16,000 x g for 5 min at 4˚C, and the resultant supernatant (1 μL) was subjected to GC-MS/MS analysis using a triple quadrupole mass spectrometer; TQ-8040 (Shimadzu Co., Kyoto Japan) with a fused silica capillary column (BPX-5; 30 m × 0.25 mm inner diameter, film thickness: 0.25 μm; Shimadzu Co.). A total of 469 metabolites covering a variety of low molecular weight small molecules such as amino acids, organic acids, and amine compounds were quantitatively monitored. The retention time and multiple reaction monitoring (MRM) conditions for those metabolites have been previously reported [19]. Triplicate samples for each medium were prepared and analyzed.

The obtained mass spectrometry peak areas for individual metabolites were normalized by the peak area of the added internal standard, 2-isopropylmalic acid. The normalized ratios were log2 transformed, metabolites that were not detected in at least two samples in at least one of the groups were removed, and missing values were imputed [20]. The log2 transformed data were then subjected to a statistical analysis using a one-way ANOVA to identify metabolites that are significantly altered in the triple species culture samples compared to the single species culture samples.

## Effect of IAA on biofilm formation

To determine the effect of IAA on biofilm formation, an *in vitro* biofilm experiment was conducted as previously described [21]. Both analysis of colony forming units (CFUs) and scanning electron microscopy (SEM) were performed. The substrate for this experiment was catheter disks (10 mm) soaked in fetal bovine serum (FBS) (Mediatech, Inc., Manassas, VA) for 24 h in a 37˚C incubator. From the frozen CT isolate, fresh plates were cultured on potato dextrose agar (PDA) containing gentamicin and chloramphenicol in a 37˚C incubator for 24 h. CT cells were then transferred to a tube containing PBS, counted using a hemocytometer, and diluted to a $1x10^7$ concentration in PBS from which 4 mL was added to the disks and allowed to adhere for 90 min at 37˚C. Following adhesion, biofilms were transferred and grown in YNB or YNB containing 50 μM IAA. Biofilms were allowed to mature for 24 h in a 37˚C incubator with rocking. Note that preliminary studies were performed to determine 50 μM of IAA as the optimal concentration. Following growth, biofilms (in triplicate) were transferred to new wells containing PBS and scraped. Next, CFUs were determined. Samples were grown on PDA plates containing antibiotics for 24 h at 37˚C. CFUs were then counted, log transformed, and analyzed.

For SEM analysis, disks were fixed in 2% glutaraldehyde for 24 h at 4˚C. Afterwards, disks were rinsed in 0.1 M sodium cacodylate buffer and transferred into 1% osmium tetroxide for 1 h (also at 4˚C). Disks were then rinsed again in 0.1 M sodium cacodylate buffer. Uranyl acetate was added to the disks and placed at 4˚C overnight. Disks were rinsed in sodium cacodylate, then twice in sterile water for 5 min. A gradual ethanol dehydration process (using 25, 50, 75, 95, and 100% ethanol) was conducted. Finally, the samples were mounted and placed into a desiccator for 48 h to complete dehydration. Afterwards, the samples were sputter-coated with

palladium and surface images were acquired using the Helios NanoLab 650 Scanning Electron Microscope.

## Effect of IAA on germination

The ability of CT to form germ tubes was assessed using a germ tube test. CT cells were cultured and diluted as described above. Then, 50 µM of IAA was added to the CT cell suspensions (5 x $10^5$ concentration). Tubes were then placed in a shaking incubator at 37˚C and 500 µL of the culture was sampled at different time points (0 h, 1 h, 3 h). The number of total cells and germinated cells were counted using a hemocytometer and percent germination was calculated.

## Isotope label-chase experiment

To each well of a 12-well plate containing a pretreated silicone catheter disk (15 mm diameter), 4 mL of the primary biofilm inoculum of CT (1 x $10^7$ cells per mL) prepared as described above was added and incubated at 37˚C for 90 min. The pretreated silicone catheter disk was prepared as described previously [21]. Nine silicone disks containing adherent cells were then removed carefully and placed gently in each well of a new 12-well plate containing 4 mL of fresh RPMI 1640 medium and incubated at 37˚C for 8 h. After the 8 h preincubation, the media from three wells were collected (0 h samples). While, the media from another six wells were replaced with $^{13}C_{11}$-Trp-RPMI 1640, and the incubation was continued another 16 h or 24 h. Following incubation, the media were collected, centrifuged at 3,000 x g for 5 min, filtered using a 0.22 µm 50 mL vacuum filter tube (MilliporeSigma, Burlington, MA), and stored until analysis.

Six Trp metabolites indole-3-lactic acid (ILA), indole-3-ethanol (tryptophol, TOL), indole-3-acetic acid (IAA), indole-3-acetonitrile (IAN), indole-3-acetamide (IAM), tryptamine (TAM)) and Trp in the culture media were analyzed as follows. Twenty microliters of $d_7$-IAA internal standard solution (1 nmol/mL) was spiked into 200 µL of each sample, then 800 µL of methanol containing 1% formic acid was added to the mixture. The sample was then centrifuged at 14,000 x g for 10 min at 4˚C. The supernatant (800 µL) was collected and dried in a Speed-vac concentrator. The residue was dissolved in 156.8 µL of 0.1% formic acid and analyzed by liquid chromatography multiple reaction monitoring (LC-MRM) as described below.

LC-MRM analysis was carried out using an Agilent 1100 HPLC system (Agilent, Santa Clara, CA) coupled with Sciex 3200 QTrap triple quadrupole mass spectrometer (Sciex, Toronto, CA) equipped with an electrospray ion source. The metabolite samples were chromatographed on a reverse-phase 2.1 x 100 mm, 2.6 µm, C18 Kinetex column (Phenomenex, Torrance, CA) using a linear gradient of acetonitrile from 2% to 80% over 8 min in aqueous 0.1% formic acid at a flow rate of 200 µL/min. The column temperature was set at 30˚C. The eluent was directly introduced into the mass spectrometer operated in the positive ion mode with multiple reaction monitoring. The ionspray voltage was set at 5,000 V, and the source temperature was set at 425˚C. The nebulizer gas, heater gas, and curtain gas supply were set at 40 psi, 40 psi, and 10 psi, respectively. Nitrogen was used as the source and collision gas. The injection volume was 10 µL. The retention time and MRM parameters for the Trp metabolites are summarized in Table 1. Representative LC-MRM chromatograms of Trp, ILA, TOL, IAA, IAN, IAM, and TAM are shown in S1 Fig.

The calibration curve consisted of six different concentrations ranging from 3.125 to 100 pmol/mL for ILA, TOL, IAA, IAM, and TAM, and from 62.5 to 20,000 pmol/mL for Trp. The $^{13}$C-labeled Trp and Trp metabolites, $^{13}C_{11}$-Trp, $^{13}C_{11}$-ILA, $^{13}C_{10}$-TOL, $^{13}C_{10}$-IAA, $^{13}C_9$-IAN, $^{13}C_9$-IAM, and $^{13}C_{10}$-TAM were quantified using the calibration curves for Trp, ILA, TOL,

**Table 1. LC-MRM parameters for Trp metabolites quantified.**

|  | R.T. (min) | MRM transition | DP (V) | CE (V) | CXP (V) |
|---|---|---|---|---|---|
| Trp | 6.1 | 205 -> 188 | 20 | 12.5 | 3 |
| $^{13}C_{11}$-Trp | 6.1 | 216 -> 199 | 20 | 12.5 | 3 |
| ILA | 8 | 206 -> 188 | 30 | 15.0 | 3 |
| $^{13}C_{11}$-ILA | 8 | 217 -> 199 | 30 | 15.0 | 3 |
| TOL | 8.6 | 162 -> 144 | 20 | 15.0 | 2 |
| $^{13}C_{10}$-TOL | 8.6 | 172 -> 154 | 20 | 15.0 | 2 |
| IAA | 8.6 | 176 -> 130 | 25 | 22.5 | 2 |
| $^{13}C_{10}$-IAA | 8.6 | 186 -> 139 | 25 | 22.5 | 2 |
| IAN | 9.7 | 157 -> 130 | 20 | 17.5 | 2 |
| $^{13}C_9$-IAN | 9.7 | 166 -> 139 | 20 | 17.5 | 2 |
| IAM | 7.5 | 175 -> 130 | 20 | 20.0 | 2 |
| $^{13}C_{10}$-IAM | 7.5 | 185 -> 139 | 20 | 20.0 | 2 |
| TAM | 6.3 | 161 -> 144 | 20 | 15.0 | 2 |
| $^{13}C_{10}$-TAM | 6.3 | 171 -> 154 | 20 | 15.0 | 2 |
| $D_7$-IAA | 8.5 | 183 -> 136 | 25 | 22.5 | 2 |

DP: declustering potential

CE: collision energy

CXP: collision cell exit potential

IAA, IAN, IAM, and TAM, respectively. The linearity of the calibration curves (peak area ratio of analyte to $d_7$-IAA versus the theoretical concentration) was verified using linear regression and accepted when the correlation coefficients of the calibration curves were greater than 0.99.

The standard stock solutions (10 μmol/mL) of ILA, TOL, IAA, IAN, IAM, and TAM were prepared in methanol. The standard stock solutions (10 μmol/mL) of Trp and $^{13}C_{11}$-Trp were prepared in 50% acetonitrile containing 0.1% formic acid. These standard stock solutions were stored at -80˚C. Calibration standard solutions were prepared fresh from the stock solutions by serial dilution with RPMI 1640 medium lacking Trp on the day of analysis. The stock solution of an internal standard ($d_7$-IAA, 10 μmol/mL) was prepared in methanol, and the working solution (1000 pmol/mL) was prepared fresh from the stock solution by serial dilution with 0.1% formic acid.

## Isotope continuous labeling experiment

The continuous labeling experiment was carried out similarly to the label-chase experiment describe above except that $^{13}C_{11}$-Trp-RPMI 1640 medium was used throughout the experiment. After the 8 h preincubation period, the culture media were changed to a fresh $^{13}C_{11}$-Trp-RPMI 1640, and the culture was continued another 16 or 24 h. Six Trp metabolites and Trp in the culture media were monitored by LC-MRM as described above.

## Results

### Indole-3-acetic acid is highly increased in the triple-species culture

Our observation that polymicrobial interactions exacerbated biofilm formation is indicative of these microbes (CT, EC and SM) interacting and communicating through secretion of small molecule metabolites [5]. Secretion of such signaling molecules is anticipated to be most robust at the initiation of the three organism co-culture. Thus, we carried out a targeted

metabolomics analysis of the single-species (CT, EC, and SM) and triple-species (CT+EC+SM) cultures to identify metabolites that are elevated in the triple-species culture. We detected a total of 284 metabolites (S1 Table). Among them, 77 metabolites were found to be most abundant in the triple-species culture, and those that were increased more than 4-fold with p-values smaller than 0.05 in the triple-species culture compared to the second most intense group are shown in Table 2. Indole-3-acetic acid (IAA) was one of the most highly increased metabolites as shown in Fig 2.

## IAA promotes biofilm formation of *C. tropicalis*

To examine the effect of IAA on biofilm formation of CT, an *in vitro* biofilm experiment was conducted. Fig 3a shows the average log CFUs ± SD for CT biofilms in the presence and absence of IAA. Our data shows that treatment with IAA resulted in a significant (p-value = 0.049) increase in average log10 CFU (5.096 ± 0.030 and 4.886 ± 0.127, respectively). SEM imaging confirmed that the addition of IAA affected the formation of CT biofilms (Fig 3b). Furthermore, in the presence of IAA an increase in the number of hyphae when compared to CT without IAA was observed (Fig 3b).

## IAA promotes germination of *C. tropicalis*

To assess the ability of CT to form germ tubes, a germ tube test was carried out. Fig 4 shows the effect of IAA on the ability of CT to germinate. As shown, following 3 h of incubation in the presence of IAA, the average percent germination of CT (35.83 ± 3.86) was significantly greater than in the absence of IAA (19.72 ± 3.85) (p-value = 0.011).

## *C. tropicalis* synthesizes IAA through the indole-3-pyruvate pathway

Given the observation that IAA promotes biofilm formation of CT, it is important to determine how IAA is produced by CT not only for understanding the mechanisms of CT biofilm

**Table 2. Metabolites highly increased in the triple species culture.**

| Metabolite.IDs | Normalized Log2 Mean Quantity | | | | | | -log10 ANOVA p-value |
|---|---|---|---|---|---|---|---|
| | Medium | CT | EC | SM | CT+EC+SM | log2 Fold-change | |
| 2-Hydroxyisocaproic acid | -11.1 | -9.4 | -8.5 | -10.6 | -4.9 | 3.6 | 13.1 |
| 2-Hydroxyisovaleric acid | -10.7 | -8.9 | -8.3 | -9.5 | -5.2 | 3.1 | 4.5 |
| 3-Aminoglutaric acid | -13.6 | -13.6 | -11.2 | -10.0 | -7.4 | 2.6 | 1.0 |
| 3-Hydroxyisobutyric acid | -14.3 | -12.6 | -14.8 | -11.5 | -6.9 | 4.7 | 2.4 |
| 3-Phenyllactic acid | -8.7 | -5.5 | -5.8 | -7.8 | -3.2 | 2.3 | 15.0 |
| 4-Hydroxyphenyllactic acid | -13.9 | -8.7 | -8.7 | -10.3 | -3.8 | 4.9 | 1.5 |
| 5-Oxoproline | -12.5 | -12.3 | -14.4 | -10.6 | 0.5 | 11.1 | 4.6 |
| Dihydrouracil | -13.5 | -12.9 | -11.8 | -11.8 | -5.7 | 6.1 | 2.9 |
| Glutaric acid | -10.9 | -10.0 | -9.2 | -12.2 | -6.4 | 2.8 | 3.4 |
| Glyceraldehyde | -13.6 | -14.3 | -14.0 | -14.5 | -4.8 | 8.9 | 6.4 |
| Indole-3-acetic acid | -9.4 | -9.4 | -9.8 | -9.2 | -5.8 | 3.4 | 8.9 |
| Kynurenic acid | -15.3 | -10.6 | -11.7 | -12.0 | -7.3 | 3.3 | 6.6 |
| Leucine | -11.5 | -9.3 | -13.3 | -14.4 | 0.8 | 10.1 | 2.2 |
| N-Acetylserine | -10.6 | -9.8 | -9.7 | -9.5 | -7.3 | 2.3 | 11.6 |
| Threitol | -8.2 | -7.2 | -8.0 | -8.4 | -5.0 | 2.1 | 10.9 |

*The fold-change is between the CT+EC+SM and the second most intense group.

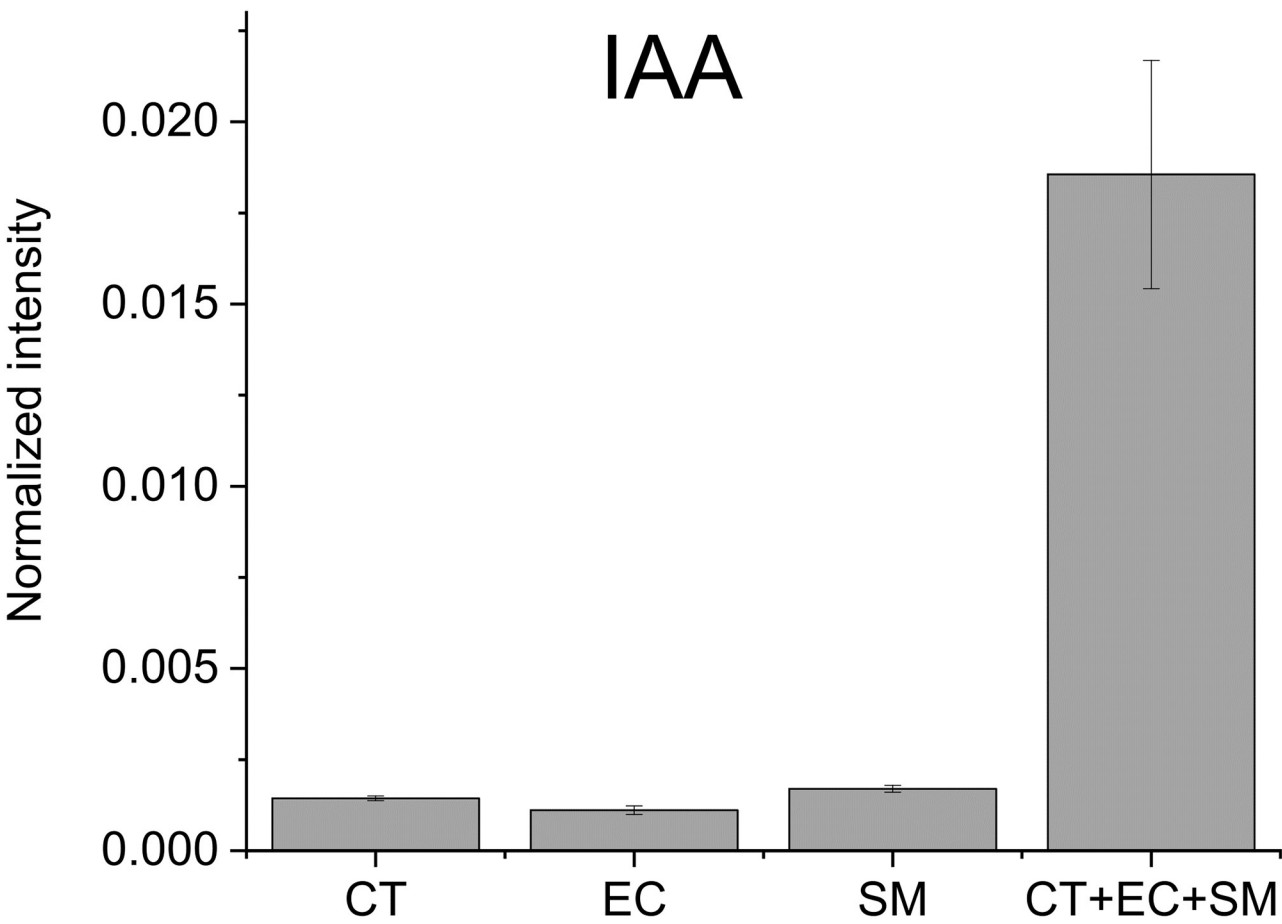

**Fig 2. Normalized abundance of IAA secreted from CT, EC, SM, and CT+EC+SM.** The results are the means from triplicate experiments and error bars indicate the means ± standard deviations.

formation, but also for developing effective means to inhibit CT biofilm formation. We investigated the biosynthesis pathway of IAA in CT by a label-chase experiment using $^{13}C_{11}$-Trp as a precursor. Trp and six Trp metabolites (ILA, TOL, IAA, IAN, IAM, and TAM) secreted into the culture medium were monitored by LC-MRM. Among the six Trp metabolites, only ILA, TOL, and IAA were observed. Since ILA can only be produced through the indole-3-pyruvate pathway, although TOL can also be produced through the tryptophan side-chain oxidation and tryptamine pathways (see Fig 1), the result indicates that the indole-3-pyruvate pathway is operating in CT to synthesize IAA.

The quantitative results on Trp and the other three metabolites (ILA, TOL, and IAA) are shown in Fig 5. As expected, only unlabeled-Trp was detected in the culture medium before switching the medium to $^{13}C_{11}$-Trp-RPMI 1640 (0 h), after which the vast majority of Trp in the culture medium became $^{13}$C-labeled Trp. In contrast, $^{13}$C-labeled ILA, TOL, and IAA were observed only after switching the medium, indicating that those metabolites were synthesized from the $^{13}$C-labeled Trp. The concentrations of those $^{13}$C-labeled metabolites secreted in the culture media were approximately 30–50 pmol/mL (30–50 nM) for ILA, 270–280 pmol/mL for TOL (270–280 nM), and 60–75 pmol/mL (60–75 nM) for IAA. The results also showed that unlabeled-ILA, TOL, and IAA were produced continuously even after switching the media to $^{13}C_{11}$-Trp-RPMI 1640, in particular, the levels of the unlabeled-ILA and IAA reached to the levels equivalent to those of $^{13}$C-labeled ILA and IAA.

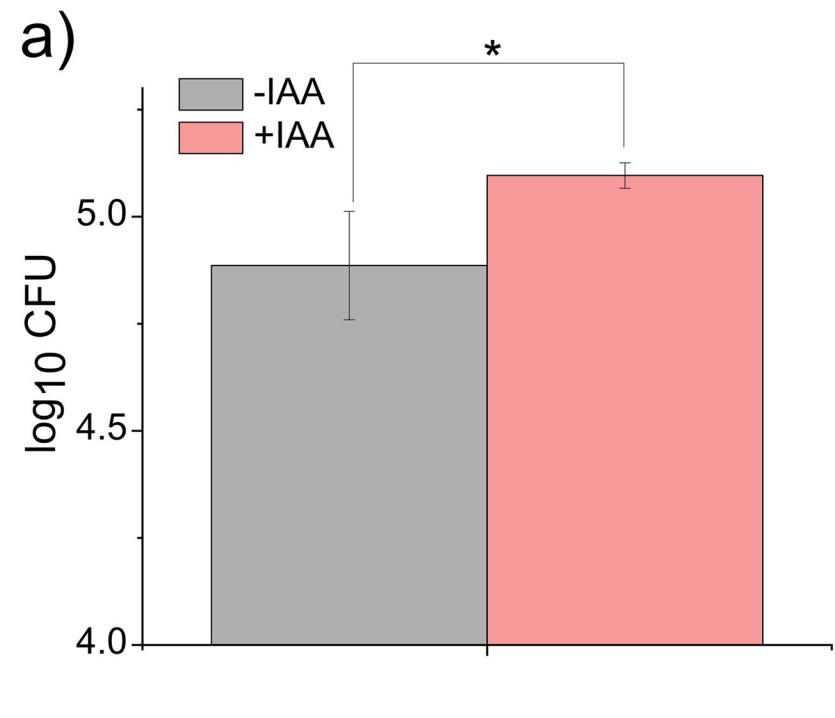

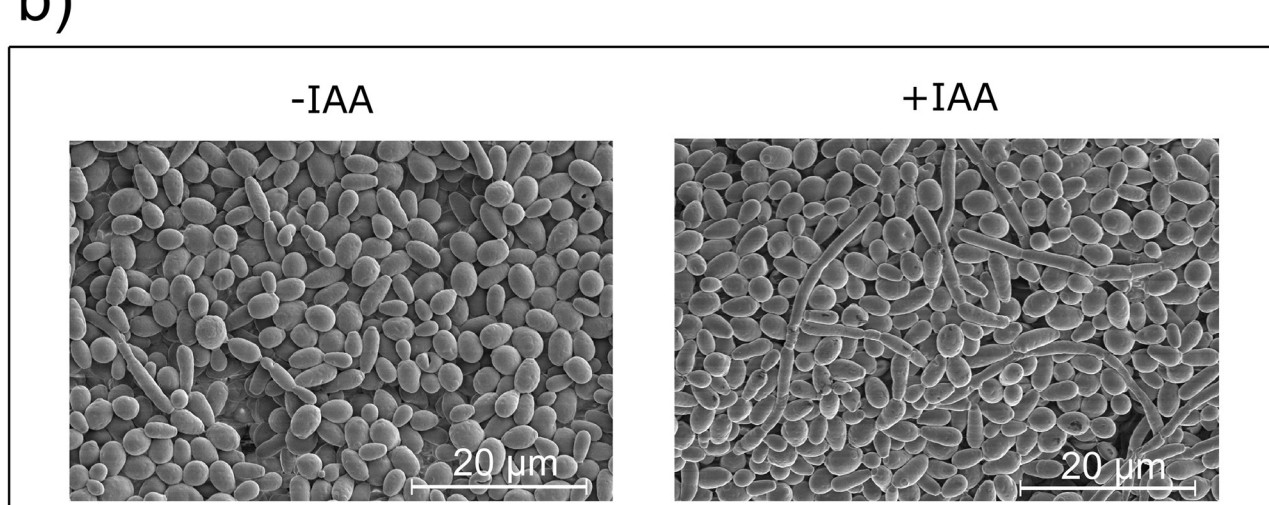

**Fig 3. Biofilm formation of *C. tropicalis* in the presence or absence of IAA.** a) *C. tropicalis* (1 x 10$^7$ cells/mL each) grown on PDA plates for 24 h in the presence or absence of IAA and colony forming units (CFUs) were counted. The results are the means from triplicate experiments and error bars indicate the means ± standard deviations. * denotes statistical significance, p-value < 0.05. b) Representative SEM images of *C. tropicalis* grown in the presence or absence of IAA, confirming that the addition of IAA promoted the formation of biofilms.

### Trp-independent pathway may also exist in *C. tropicalis*

To determine how much of ILA, TOL, and IAA were synthesized from Trp supplemented in the culture medium, CT was grown in $^{13}C_{11}$-Trp-RPMI 1640 media throughout the experiment, and the levels of the $^{13}C$-labeled and unlabeled metabolites in the culture medium were measured. The continuous labeling experiment showed that about 60% of ILA, 20% of TOL, and 50% of IAA produced were in the unlabeled form (Fig 6). Interestingly, 300–500 pmol/mL

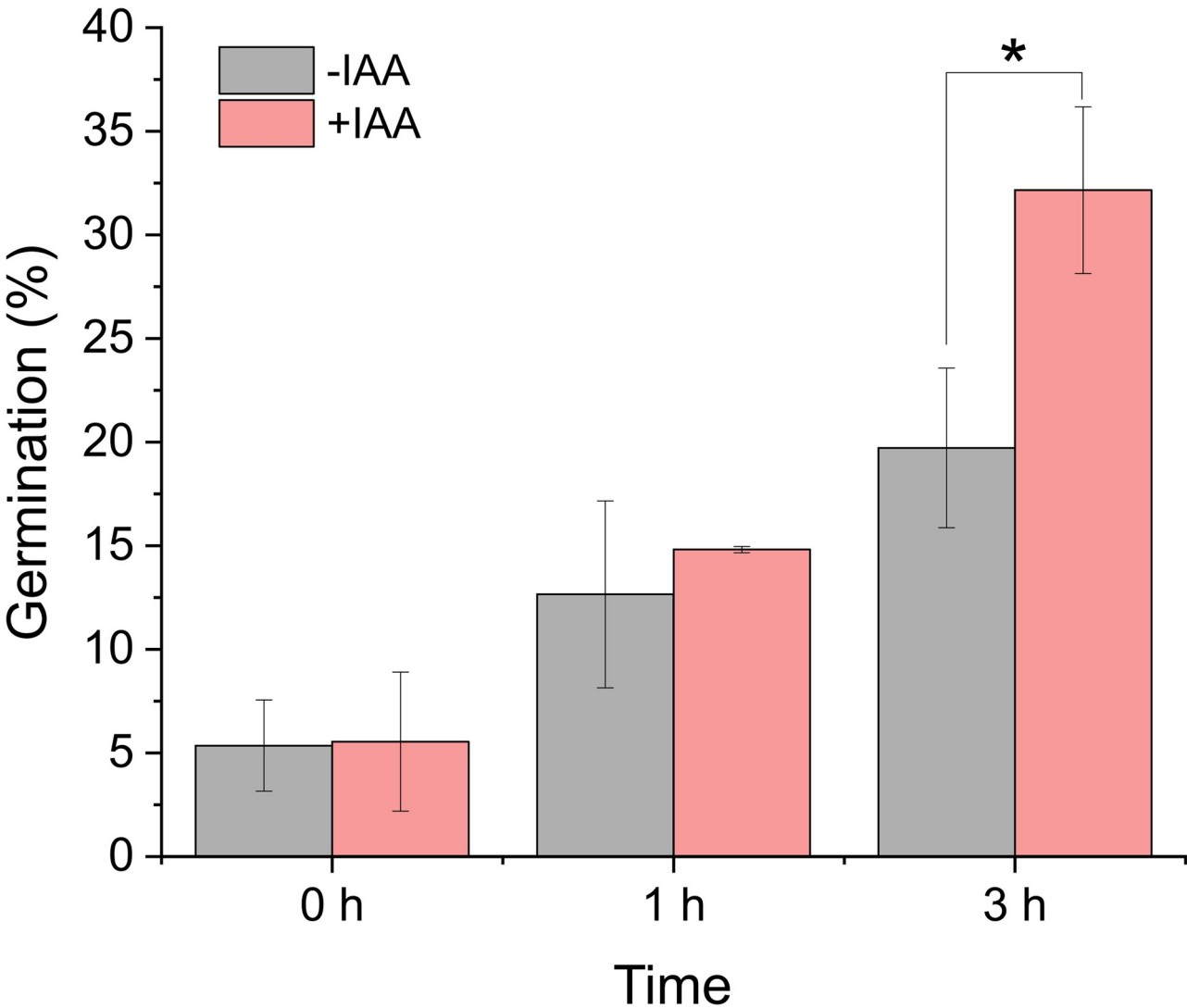

**Fig 4. Extent of germination of *C. tropicalis* in the presence or absence of IAA.** *C. tropicalis* (0.3 x 107 cells/mL each) was grown in PBS for 0, 1, or 3 h in the presence or absence of IAA. At the indicated time points, an aliquot was taken and the numbers of total cells and germinated cells were counted. The results are the means from triplicate experiments and error bars indicate the means ± standard deviations. * denotes statistical significance, p-value < 0.05.

(300–500 nM) of unlabeled Trp was clearly observed after 16 and 24 h of incubation, suggesting that CT may be able to synthesize Trp.

## Discussion

We demonstrated that IAA promotes biofilm formation as well as germination of *C. tropicalis* (Figs 3 and 4). Prior reports also showed that IAA induces the hyphal formation of *Candida albicans* [15], *Tricholoma vaccinum* [13], and *Saccharomyces cerevisiae* [16]. Thus, the hyphal induction effect of IAA may be common to many fungal species. Although there are many environmental cues that have been shown to induce the dimorphic (a yeast (or yeast-like) phase and a mold (filamentous) phase) switching of fungi, the cell signaling pathways central to dimorphic switching appeared to be well conserved [22]. Signaling pathways associated

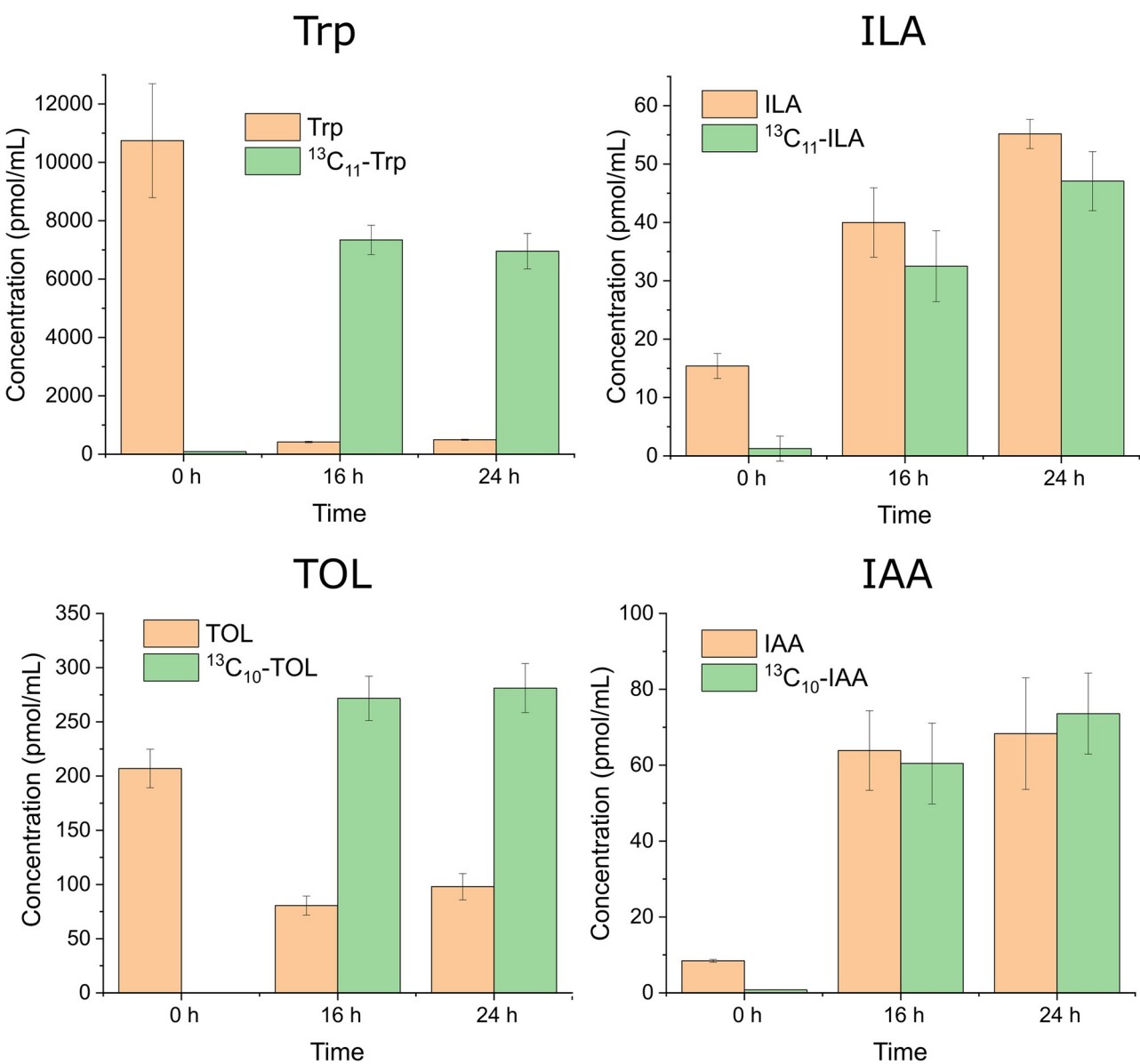

**Fig 5. Production of ¹³C-labeled Trp, ILA, TOL and IAA before and after switching the medium to $^{13}C_{11}$-Trp-RPMI 1640.** *C. tropicalis* was initially cultured in a regular RPMI 1640 for 8 h (0 h) and the culture medium was changed to an $^{13}C_{11}$-Trp-RPMI 1640 medium and the culture was continued another 16 or 24 h. The results are the means from triplicate experiments and error bars indicate the means ± standard deviations.

with dimorphic switching are the mitogen-activated protein kinase (MAPK) and the cAMP-dependent protein kinase A (PKA) pathways. Rao and co-workers tested whether IAA induced hyphal formation involves the MAPK or PKA pathway using mutant strains of *C. albicans* defective in the MAPK and/or PKA pathways. Interestingly, IAA-induced hyphal formation occurred in those mutants, suggesting that IAA-mediated filamentation occurs via a MAPK and PKA independent pathway [15]. Understanding the mechanism of IAA-induced hyphal formation would help modulate the formation of fungal biofilm.

The isotope tracing experiments demonstrated that *C. tropicalis* synthesizes IAA through the indole-3-pyruvate pathway. This was concluded by the observation of ¹³C-labeled ILA,

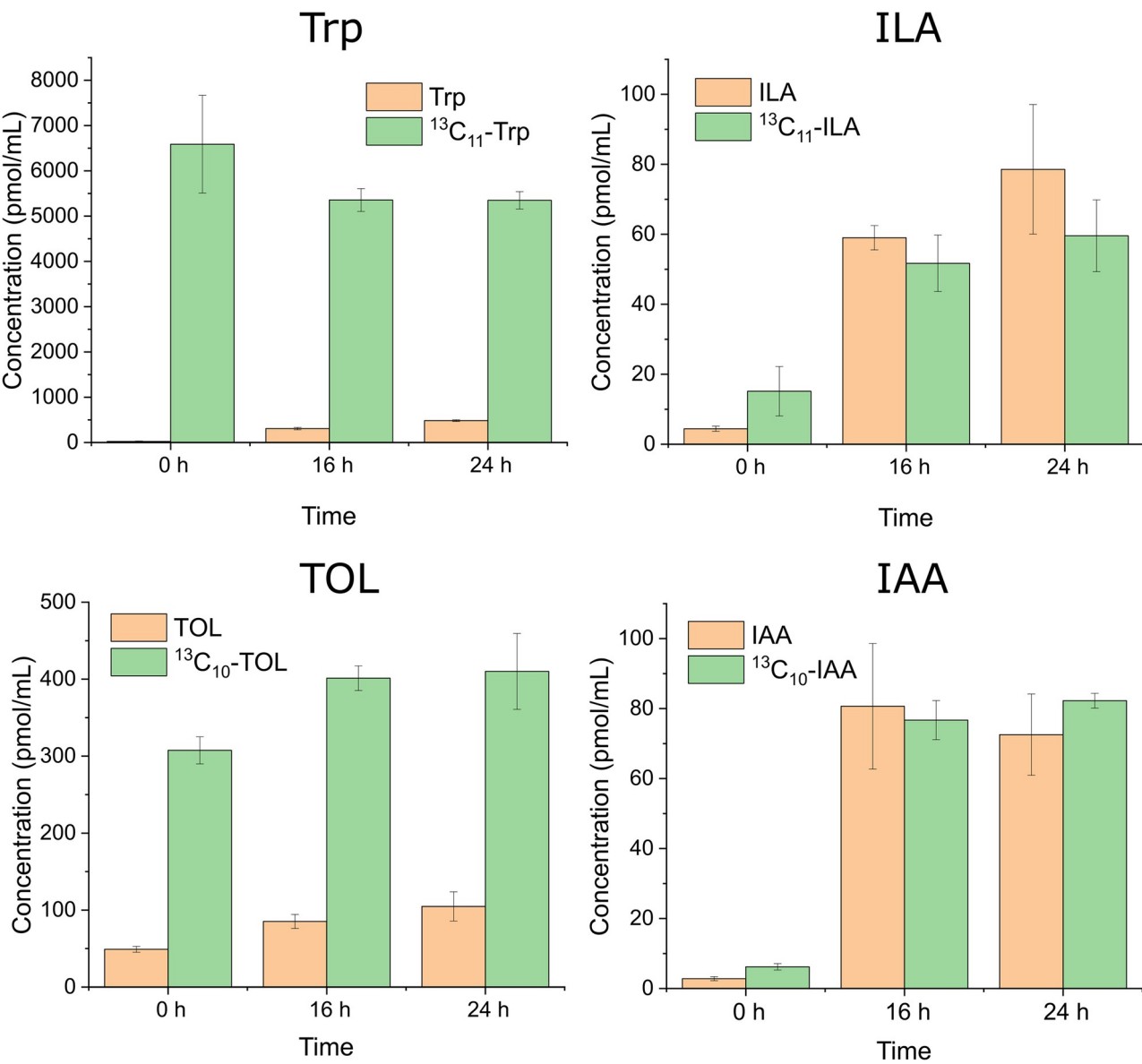

**Fig 6. Production of $^{13}$C-labeled Trp, ILA, TOL and IAA when *C. tropicalis* was cultured continuously with $^{13}$C$_{11}$-Trp-RPMI 1640 medium.** After the 8 h pre-incubation period (0 h), the culture medium was changed to a fresh $^{13}$C$_{11}$-Trp-RPMI 1640 and the culture was continued another 16 or 24 h. The results are the means from triplicate experiments and error bars indicate the means ± standard deviations.

TOL, and IAA (Fig 5) because ILA can be produced only through the indole-3-pyruvate pathway and all the three metabolites are produced through the indole-3-pyruvate pathway (Fig 1). The presence of genes coding the three enzymes involved in the indole-3-pyruvate pathway in the *C. tropicalis* genome (ATCC MYA-3404) also supports our conclusion. These are genes for tryptophan aminotransferase (*CTRG_01079*), which converts Trp to indole-3-pyruvate, pyruvate decarboxylase (*CTRG_02361*, *CTRG_00232*, and *CTRG_03826*), which converts indole-3-pyruvate to indole-3-acetaldehyde, and indole-3-acetaldehyde dehydrogenase (*CTRG_01342*), which converts indole-3-acetaldehyde to indole-3-acetic acid. It should also be noted that the pathway that skips the formation of indole-3-acetaldehyde and directly synthesizes IAA from indole-3-pyruvic acid may exist in *C. tropicalis* because a putative

flavin monooxygenase gene, which converts indole-3-pyruvic acid directly to IAA, has been identified in the *C. tropicalis* genome [23]. Furthermore, it is possible that the tryptophan side-chain oxidation pathway, in which Trp is directly converted to indole-3-acetaldehyde and then to IAA, may also exist. However, the enzyme that carries out the catalysis, tryptophan side-chain oxidase, has only been demonstrated in *Pseudomonas fluorescen* [24]. As far as we know, this is the first study characterizing the biosynthesis pathway of *Candida* species. The outcome of the study advances our knowledge regarding the potential for investigating IAA on the formation of the mixed-species biofilm using inhibitors to the enzymes or knocking out the genes involved in the pathway.

Our continuous labeling experiment suggested that a Trp-independent pathway may also exist in *C. tropicalis*. The continuous labeling experiment revealed that about 60% of ILA, 20% of TOL, and 50% of IAA produced were not derived from Trp supplemented in the culture medium (Fig 6). It seems plausible that two mechanisms may explain the observation. One possible mechanism would be the involvement of a Trp-independent pathway. The existence of the Trp-independent pathway has been suggested in yeast strains [25]. Therefore, the Trp-independent pathway could commonly exist in fungi; however, no enzymes or genes involved have been identified thus far. Another possible mechanism would be that unlabeled-Trp was *de novo* synthesized and then fed to the indole-3-pyruvate pathway. The enzyme that catalyzes the last two steps in the biosynthesis of Trp, tryptophan synthase, is widely expressed in bacteria, plants, and fungi [26]. Indeed, a gene encoding tryptophan synthase (*CTRG_04034*) exists in the *C. tropicalis* genome. Furthermore, we observed *de novo* synthesized unlabeled Trp after 16 and 24 h of incubation (Fig 6). Thus, it is highly likely that the continuous production of unlabeled ILA, TOL and IAA are due to the activity of *de novo* production of unlabeled Trp. However, further investigations are likely to be necessary to determine the exact mechanism(s) supporting this observation.

The metabolomics analysis of the single-species and triple-species cultures of CT, EC and SM identified 15 metabolites that were highly increased in the triple-species culture, including IAA (Table 2). Further study is required to evaluate the effects of those molecules on the biofilm formation of CT. The analysis demonstrates that metabolomics is useful to identify candidate molecules that may be associated with biofilm formation.

## Supporting information

**S1 Fig. Representative overlaid MRM chromatograms of Trp and six Trp metabolites.** The concentrations of the metabolites are 1 nmol/mL.
(TIF)

**S1 Table. Metabolites identified.**
(XLSX)

## Acknowledgments

The authors wish to thank Yu-Shiuan Cheng for assisting mass spectroscopy analysis of tryptophan metabolites.

## Author Contributions

**Conceptualization:** Masaru Miyagi, Thomas McCormick, Mahmoud A. Ghannoum.

**Formal analysis:** Masaru Miyagi, Rachel Wilson.

**Funding acquisition:** Mahmoud A. Ghannoum.

**Investigation:** Masaru Miyagi, Rachel Wilson, Daisuke Saigusa, Keiko Umeda, Reina Saijo, Christopher L. Hager, Yuejin Li.

**Methodology:** Masaru Miyagi, Daisuke Saigusa.

**Supervision:** Mahmoud A. Ghannoum.

**Writing – original draft:** Masaru Miyagi.

**Writing – review & editing:** Thomas McCormick, Mahmoud A. Ghannoum.

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
