## [Decision Letter · Decision Letter 0]

23 Nov 2020

PONE-D-20-28447

Indole-3-acetic acid synthesized through the indole-3-pyruvate pathway promotes Candida tropicalis biofilm formation

PLOS ONE

Dear Dr. Miyagi,

Thank you for submitting your manuscript to PLOS ONE. After careful consideration, we feel that it has merit but does not fully meet PLOS ONE’s publication criteria as it currently stands. Therefore, we invite you to submit a revised version of the manuscript that addresses the points raised during the review process.

We look forward to receiving your revised manuscript.

Kind regards,

Rashid Nazir

Academic Editor

PLOS ONE

Journal Requirements:

2. Please upload a copy of Figure 6, to which you refer in your text on page 19. If the figure is no longer to be included as part of the submission please remove all reference to it within the text.

Reviewers' comments:

Reviewer's Responses to Questions

**Comments to the Author**

1. Is the manuscript technically sound, and do the data support the conclusions?

Reviewer #1: Partly

2. Has the statistical analysis been performed appropriately and rigorously? 

Reviewer #1: Yes

3. Have the authors made all data underlying the findings in their manuscript fully available?

Reviewer #1: Yes

4. Is the manuscript presented in an intelligible fashion and written in standard English?

Reviewer #1: Yes

5. Review Comments to the Author

Reviewer #1: In the manuscript entitled “Indole-3-acetic acid synthesized through the indole-3-pyruvate pathway Promotes Candida tropicalis biofilm formation”, Miyagi and coworkers discovered indole-3-acetic acid is a small signaling molecule, which can promote the biofilm formation of Candida tropicalis. This work is interesting but there are some issues needed to be addressed before accepted for publication.

1. In page 6 line 117. a space was missing in 2500g.

2. The authors should carefully check the units throughout the manuscript, for example, hours/h, minute/min. The format should be uniform.

3. Another format issues, for example, Five-hundred µL of the culture; 250 μL of mixed water:methanol:chloroform. Please use the arabic numerals throughout the manuscript.

4. In result section, in part 1 (page 13), the author identified 15 overexpression metabolites from 284 metabolites. But in part 2 (page 14), the author directly claimed that IAA promotes biofilm formation of C. tropicalis. I wonder the other 14 metabolites if possess the similar function? It is better to provide the corresponding evidence.

5. In page 24, Figure 4 caption, “0.3 x 107 cells/mL” “7” should be displayed in superscript.

6. PLOS authors have the option to publish the peer review history of their article (what does this mean?). If published, this will include your full peer review and any attached files.

Reviewer #1: **Yes: **Dongliang Yang

---

## [Author Response · Author response to Decision Letter 0]

23 Nov 2020

Dear Dr. Nazir

We have revised our manuscript “Indole-3-acetic acid synthesized through the indole-3-pyruvate pathway promotes Candida tropicalis biofilm formation” by Miyagi et al. for consideration for publication in Plos One. We appreciate your invitation to resubmit the manuscript for another review. The reviewer’s suggestions/comments have improved the manuscript quality, and we appreciate the reviewer’s insight. The changes we have made are solely in the main text, and all the figures and tables remain the same. Fig. 6, which was not uploaded in the original submission, has been submitted. Below are our responses to the reviewer’s comments. We hope that the manuscript is now acceptable for publication in Plos One. We are looking forward to hearing your decision.

Responses to Reviewer 1:

Comment 1: In page 6 line 117. a space was missing in 2500g.

A space has been added.

Comment 2: The authors should carefully check the units throughout the manuscript, for example, hours/h, minute/min. The format should be uniform.

The units for hour and min have been changed to “h” and “min,” respectively. They are now uniformly used throughout the manuscript.

Comment 3: Another format issues, for example, Five-hundred µL of the culture; 250 μL of mixed water:methanol:chloroform. Please use the arabic numerals throughout the manuscript.

Arabic numerals are now used for all the numbers that are followed by a unit of measurement.

Comment 4: In result section, in part 1 (page 13), the author identified 15 overexpression metabolites from 284 metabolites. But in part 2 (page 14), the author directly claimed that IAA promotes biofilm formation of C. tropicalis. I wonder the other 14 metabolites if possess the similar function? It is better to provide the corresponding evidence.

 We agree with the reviewer that it is interesting to investigate whether the other 14 metabolites affect the formation of biofilm. We chose IAA among 15 metabolites because it has been indicated to induce filamentation of certain fungi. We are currently investigating the other 14 metabolites. However, this manuscript is focused on IAA, and extending the study to other metabolites is beyond this report's scope.

Comment 5: In page 24, Figure 4 caption, “0.3 x 107 cells/mL” “7” should be displayed in superscript.

 It has been corrected.

Sincerely,

---

## [Decision Letter · Decision Letter 1]

7 Dec 2020

Indole-3-acetic acid synthesized through the indole-3-pyruvate pathway promotes Candida tropicalis biofilm formation

PONE-D-20-28447R1

Dear Dr. Miyagi,

We’re pleased to inform you that your manuscript has been judged scientifically suitable for publication and will be formally accepted for publication once it meets all outstanding technical requirements.

Kind regards,

Rashid Nazir

Academic Editor

PLOS ONE

Additional Editor Comments (optional):

Reviewers' comments:

Reviewer's Responses to Questions

**Comments to the Author**

1. If the authors have adequately addressed your comments raised in a previous round of review and you feel that this manuscript is now acceptable for publication, you may indicate that here to bypass the “Comments to the Author” section, enter your conflict of interest statement in the “Confidential to Editor” section, and submit your "Accept" recommendation.

Reviewer #1: All comments have been addressed

2. Is the manuscript technically sound, and do the data support the conclusions?

Reviewer #1: Partly

3. Has the statistical analysis been performed appropriately and rigorously? 

Reviewer #1: Yes

4. Have the authors made all data underlying the findings in their manuscript fully available?

Reviewer #1: Yes

5. Is the manuscript presented in an intelligible fashion and written in standard English?

Reviewer #1: Yes

6. Review Comments to the Author

Reviewer #1: (No Response)

7. PLOS authors have the option to publish the peer review history of their article (what does this mean?). If published, this will include your full peer review and any attached files.

Reviewer #1: No

---

## [Editor Report · Acceptance letter]

9 Dec 2020

PONE-D-20-28447R1 

Indole-3-acetic acid synthesized through the indole-3-pyruvate pathway promotes *Candida tropicalis* biofilm formation 

Dear Dr. Miyagi:

I'm pleased to inform you that your manuscript has been deemed suitable for publication in PLOS ONE. Congratulations! Your manuscript is now with our production department. 

Kind regards, 

on behalf of

Dr Rashid Nazir 

Academic Editor

PLOS ONE